# Reduced Graphene Oxide/Carbon Paper for the Anode Diffusion Layer of a Micro Direct Methanol Fuel Cell

**DOI:** 10.3390/nano12172941

**Published:** 2022-08-26

**Authors:** Dacheng Zhang, Kang Li, Ziten Wang, Zhengang Zhao

**Affiliations:** 1Faculty of Information Engineering and Automation, Kunming University of Science and Technology, Kunming 650500, China; 2Yunnan Key Laboratory of Computer Technologies Application, Kunming 650500, China; 3Yunnan Key Laboratory of Green Energy, Electric Power Measurement Digitalization, Control and Protection, Kunming 650500, China

**Keywords:** micro direct methanol fuel cell, reduced graphene oxide, anode diffusion layer, methanol crossover

## Abstract

The diffusion layer (DL) in the structure of the membrane electrode assembly (MEA) of a micro direct methanol fuel cell (*μ*DMFC) plays an essential role in reactant/product mass transfer and catalyst loading. The material selection and structure design of the *μ*DMFC affects its performance. In this work, a reduced graphene oxide/carbon paper (rGO/CP) was proposed and prepared for the anode DL of a *μ*DMFC. It was prepared using electrophoretic sedimentation combined with an in situ reduction method. The rGO/CP reduced the cell’s ohmic and charge transfer resistances. Meanwhile, it provided more significant mass transfer resistance to reduce the methanol crossover, allowing the cell to operate stably at higher concentrations for a longer duration than conventional *μ*DMFCs. The experimental results showed that the maximum power density increased by 53% compared with the traditional anode DL of carbon paper.

## 1. Introduction

A micro direct methanol fuel cell (μDMFC) is a miniature electrochemical energy conversion device. It has been considered as a potential alternative to lithium-ion batteries thanks to its high power density, accessible storage, simple structure, etc., which make it suitable for use in portable electronics and vehicle applications [1,2,3,4,5]. Its essential part, the membrane electrode assembly (MEA), consists of an anode electrode, a cathode electrode, and a membrane sandwiched between these parts [6]. Both electrodes contain a diffusion layer (DL) and a catalytic layer (CL). The DL surface is covered with a thin microporous carbon nanoparticle layer that helps to minimize catalyst losses [7]. Although the DL does not participate directly in the electrochemical reaction, it is responsible for controlling the mass transfer of reactants and products [8]. Moreover, it provides a support structure for the CL and connects it to the collector; therefore, the DL ought to be appropriately porous, highly conductive, chemically stable, and of excellent rigidity [9]. Carbon paper (CP) and carbon cloth (CC) are the most widely used materials for DLs in μDMFCs.

The performance of a μDMFC can be below the expected theoretical value due to various losses, such as activation loss, ohmic loss, mass transfer loss, methanol crossover, etc. As an essential structure in the MEA, an appropriate DL design can effectively diminish those adverse effects and improve cell performance. Xue et al. [10] deposited reduced graphene oxide in some stainless steel fiber felt for use as a gas diffusion layer (GDL) and a cathode current collector, which improved the reverse water diffusion, especially at high concentrations. Braz et al. [11] studied the effects of the characteristics of an anode DL on the performance of a passive DMFC. A better performance was obtained when using bilayer structures for the anode DL and current collector rather than CPs. Zhu et al. [12] used three-dimensional graphene (3DG) for the gas DL of a DMFC, which can provide a low contact resistance and a sufficient fuel diffusion path. Zhu et al. [13] designed a new button-type μDMFC with a 3DG GDL and a built-in spring. It was found that this μDMFC had a lower ohmic resistance and mass transfer resistance than conventional ones. Abdelkareem et al. [14] found that the utilization of CP by the catalyst was higher than that of CC. Yuan et al. [15] used 3DG to construct a cathode MPL, whose inherent pore structure and characteristics enhanced water management; it reduced the water transport coefficient and methanol crossover. Yan et al. [16] created large methanol concentration gradients by reducing the porosity of the anode diffusion layer using nano-carbon powder. The results showed that using the modified CP for an anode DL enabled the cell to work at concentrations of up to 10 mol/L without sacrificing its performance. Alrashidi et al. [17] obtained a new ADL, with both hydrophobic and hydrophilic paths, by laser perforation of a PTFE-treated anode DL. It provided convenient transport paths for liquid methanol solutions and CO_2_; however, perforation increased methanol crossover, and if the perforation density of the abode DL was too high, the methanol crossover would be aggravated.

The physical properties of a DL, as a reactant/product transfer channel and a support structure for the CL, have a very profound effect on the electrochemical reactions in the cell [14,18], such as gas–liquid two-phase mass transfer [10,12,13,15,17], ohmic polarization [12,13,19], and methanol crossover [10,15,16,17]. Graphene is highly conductive and mechanically and chemically stable; thus, it is widely used in DLs in μDMFCs [10,12,13,15,20]. However, its processability and manufacturing costs are the main obstacles to its large-scale commercialization [12]. In this work, graphene oxide was reductively deposited onto carbon paper using electrophoretic sedimentation combined with in situ reductions to reduce its processing difficulty and manufacturing cost. The carboxyl groups at the edge of the graphene oxide dissociated under the alkaline environment to form negatively-charged suspended particles. These migrated towards the anode and were loaded on to the surface of the carbon paper under the action of the applied electric field. Electrons were then released to form RCOO-radicals, which further decarboxylated to form R-radicals. Subsequently, the radicals polymerized with each other to form covalent bonds in order to complete the reduction of the graphene oxide [21]. In this study, after a systematic study of the rGO/CP in terms of physical morphology, wettability, and porosity, a novel μDMFC was produced with an rGO/CP anode DL. Its performance was compared with that of a conventional μDMFC with a CP anode DL. The effects of the rGO/CP on activation and ohmic and concentration polarization were studied using an electrochemical impedance spectroscopy (EIS) technique [22,23,24]. The effect of the rGO/CP on methanol crossover was investigated by applying a linear sweep voltammetry (LSV) technique [17].

## 2. Materials and Methods

### 2.1. Reduced Graphene Oxide/Carbon Paper Preparation

A total of 0.01 mol/L borax standard buffer solution was prepared, and 2 mg/mL graphene oxide solid powder (Knano Graphene Technology, Xiamen, China) and 0.1 mol/L Na_2_SO_4_ were then added. The solution was then ultrasonically dispersed for two hours to sufficiently mix the solution for the graphene oxide suspension. The pool voltage was controlled by a DC power supply. The platinum electrode, carbon paper TGPH-060 (Toray, Tokyo, Japan), and graphene oxide suspension were then used for the cathode, anode, and electrolyte, respectively. The CP was deposited electrophoretically at 15 V for 10 min, cleaned ultrasonically, and dried at 100 °C for two hours to obtain the rGO/CP [25,26], as shown in Figure 1.

The surface morphology was examined using a scanning electron microscopy Zeiss Sigma 300 (Zeiss Group, Oberkochen, Germany). The wettability was investigated using a contact angle tester JY-82C (Dingsheng Testing Equipment, Chengde, China). The porosity was measured using a mercury porosimeter AutoPore Iv 9510 (Micromeritics, Norcross, GA, USA).

### 2.2. Single Cell Preparation

Two MEAs, with different anode DLs of CP and rGO/CP, respectively, were prepared for comparison. Nafion 117 was pretreated with 3 wt.% H_2_O_2_, 3 wt.% H_2_SO_4_, and deionized water at 80 °C for one hour in each solution in turn [10,27]. Then, the carbon powder, 10 wt.% PTFE solution, and ethylene glycol were mixed into a slurry, which was brushed onto the pretreated CP and rGO/CP as a micro porous layer (MPL) [7]. It was then put into a vacuum tube furnace to sinter at 340 °C in order to obtain the porous structure. The anode catalyst solution was made by mixing deionized water, isopropanol, and 5 wt.% Nafion solution with PtRu/C (30 wt.% Pt, 30 wt.% Ru, 40 wt.% C) and sprayed uniformly on to the previously prepared MPL as the anode catalytic layer. The PtRu was loaded on the anode catalytic layer at a rate of 4 mg/cm^2^. The cathode catalytic layer was prepared the same way as the anode, except that Pt/C (40 wt.% Pt, 60 wt.% C) was used as the catalyst, and the Pt was loaded on to the cathode catalytic layer at a rate of 2 mg/cm^2^. Finally, MEAs with an effective area of 1 cm × 1 cm were formed by sandwiching Nafion 117 between the gas diffusion electrodes and hot pressed at 408 K, 1 MPa for four minutes. The μDMFC consisted of the end plates, current collectors, MEA, and gaskets on both the cathode and anode sides, as shown in Figure 2.

The current collectors on both sides were made from hole-type 304 stainless steel with 38.5% opening ratios and 1 mm thicknesses [28]. A 1.2 cm × 1.2 cm × 1.5 cm fuel chamber was provided on the anode end plate to store methanol. This work required the preparation of two different μDMFCs: a CP-μDMFC and an rGO/CP-μDMFC.

### 2.3. Electrochemical Tests

After the cells were assembled, a gradient discharge activated them, and testing began. The testing platform consisted of a DC electronic load T8511A+ (ITECH, New Taipei City, Taiwan), an electrochemical workstation CHI660E (CH Instruments, Austin, TX, USA), and a thermostat 101-0A (SAIDELISI, Tianjing, China). All the following tests were performed at the operating temperature of 343 K.

#### 2.3.1. Performance Measurements

The DC electronic load gradually increased the current in 5 mA intervals to obtain the cell’s current–voltage and current–power curves at different methanol concentrations. The cell was allowed to discharge under $80\;{\mathrm{mA}/\mathrm{cm}}^{2}$ until the voltage dropped to 0 V. The curve of the voltage variation with time was recorded.

#### 2.3.2. Electrochemical Impedance Spectroscopy (EIS)

The cell was set to a stable discharge of 80 mA/cm^2^, the frequency range of the sinusoidal voltage was set from 100 KHz to 0.01 Hz, and the frequency range was divided into seven intervals: 100 KHz∼10 KHz, 10 KHz∼1 KHz, 1 KHz∼100 Hz, 100 Hz∼10 Hz, 10 Hz∼1 Hz, 1 Hz∼0.1 Hz, 0.1 Hz∼0.01 Hz, and eight points were taken in each interval.

#### 2.3.3. Linear Sweep Voltammetry (LSV)

The cell cathode was sealed and discharged at a constant current density of 100 mA/cm^2^ until the voltage dropped to 0 V to consume the remaining oxygen at the cathode side. Then, a linear scan was performed from 0 V to 1.8 V at a scan rate of 50 mV/s.

### 2.4. Methanol Mass Transfer Analysis

In a passive DMFC, the flux of methanol from the fuel chamber through the ADL to the ACL can be expressed as follows [16,29]:(1)$$N=-D_{ADL}\nabla C_{m}$$
where $D_{ADL}$ is the effective diffusivity of methanol in the ADL and $C_{m}$ is the methanol concentration.

In porous materials, the effective diffusivity of methanol is not exactly equivalent to the liquid-phase methanol diffusivity and requires a correction [30]:(2)$$D_{ADL}=\frac{\varepsilon_{ADL}}{\tau }D$$
where $\varepsilon_{ADL}$ is the porosity of the ADL, τ is the tortuosity, and *D* is the liquid-phase methanol diffusivity.

There is a standard correlation between tortuosity and porosity, according to the Bruggeman model [30]:(3)$$\tau =\varepsilon_{ADL}^{-0.5}$$

Therefore, the effective diffusivity of methanol in the ADL is corrected by applying the Bruggeman model:(4)$$D_{ADL}=\varepsilon_{ADL}^{1.5}D$$

In the ACL, part of the methanol is consumed by a methanol oxidation reaction (MOR) and the rate of methanol consumption is as follows [16,29]:(5)$$N_{OR}=\frac{i}{6F}$$
where *i* is the current density and *F* is Faraday’s constant.

Under steady-state conditions we have the following [16,29]:(6)$$N=N_{OR}+N_{cross}$$
where $N_{cross}$ is the methanol crossover flux.

The diffusion flux from the fuel chamber is the only source of methanol. Meanwhile, $N_{OR}$ only depends on the current density value. It can decrease the methanol diffusion flux by increasing the methanol mass transfer resistance in the ADL to reduce the methanol crossover.

## 3. Results and Discussion

### 3.1. Physical Characterization

We investigated the changes in the physical characterizations of the rGO/CP compared with the CP. Figure 3 shows the SEM images of the CP and rGO/CP, and Figure 4 shows the contact angles.

As shown in Figure 3a,c,e, the untreated CP was composed of smooth carbon fibers with diameters of 8∼10 μm. In Figure 3b,d,f, after the CP was electrophoretically deposited, the carbon fibers were attached with a cluster-like substance. The comparison showed that the electrophoretic sedimentation combined with the in situ reduction method could effectively load rGO on to the surface of the carbon fibers. To investigate the effect of rGO on the porosity of the CP we used mercury intrusion porosimetry (MIP) to quantify the porosity of the rGO/CP (70.91%), which was 78% lower than that of CP [11,31]. According to Equations (Equation 1) and (Equation 4)–(Equation 6), it could be concluded that reducing the porosity of ADL ($\varepsilon_{ADL}$) could reduce the flux of the methanol crossover, $N_{cross}$.

As shown in Figure 4, the CP had a contact angle of $119.59^{\circ }$ with the water droplets, indicating that the surface was hydrophobic. The rGO/CP had a contact angle of $80.51^{\circ }$ with the water droplets, indicating that the surface was hydrophilic. However, compared with the CP, the hydrophilic rGO/CP could not remove the CO_2_ generated by the MOR. Instead, it increased the mass transfer resistance of the methanol solution from the fuel chamber to the ACL due to its smaller contact angle, which reduced the methanol crossover. Sun et al. [32] found that a higher ADL contact angle usually led to increased methanol crossover. Although it increased the methanol flow resistance at the inlet, it reduced the mass transfer resistance at the ADL/AMPL interface, and the capillary-driven flow was also enhanced inside the ADL. Therefore, the rGO/CP with a smaller contact angle would reduce the methanol crossover. Thus, using rGO/CP with a lower porosity and a smaller contact angle for an ADL would result in a reduction in methanol crossover and help the μDMFC to increase the methanol feeding concentration. Moreover, the excellent conductivity of rGO would also reduce the ohmic resistance of the cell and improve the power density of the μDMFC.

### 3.2. μDMFC Performance

In order to investigate the effect of the rGO/CP on the power density of the μDMFC, the polarization curves of the CP-μDMFC and the rGO/CP-μDMFC at different methanol solution concentrations (0.5 mol/L∼5.0 mol/L) were tested using a 343 K thermostat. Figure 5 shows the polarization curves of the CP-μDMFC and the rGO/CP-μDMFC at different methanol solution concentrations.

As shown in Figure 5a, the CP-μDMFC obtained its maximum power density at 1 mol/L, which was 23.36 mW/cm^2^. The power density of the CP-μDMFC decreased abruptly at high current densities with lower concentrations (0.5 mol/L, 1 mol/L), which was due to the rapid rate of methanol consumption at the high current densities, resulting in a poor fuel supply. At higher concentrations (3 mol/L, 4 mol/L, and 5 mol/L), the aggravation of the methanol crossover would lead to a poor cell performance. As shown in Figure 5b, the rGO/CP-μDMFC obtained the maximum power density at 3 mol/L, which was 35.72 mW/cm^2^. Compared with the CP-μDMFC, it had an improved optimal methanol feeding concentration and maximum power density. The optimal methanol feeding concentration was enhanced from 1 mol/L to 3 mol/L, and the maximum power density was improved from 23.36 mW/cm^2^ to 35.72 mW/cm^2^. Moreover, the open-circuit voltage and maximum operating current of the rGO/CP-μDMFC at higher concentrations were higher than those of the CP-μDMFC. With an increasing methanol concentration, the improvement of the rGO/CP-μDMFC became significant; this implied that the ADL of the rGO/CP could effectively relieve methanol crossover, especially at higher concentrations.

The maximum power density and open-circuit voltage of the CP-μDMFC and the rGO/CP-μDMFC at different methanol concentrations are listed in Table 1.

It can be seen in Table 1 that the rGO/CP-μDMFC outperformed the CP-μDMFC at all tested concentrations as the addition of rGO reduced the ohmic resistance and charge transfer resistance of the rGO/CP-μDMFC and enhanced the cell’s performance. The optimal methanol feeding concentration of the rGO/CP-μDMFC was enhanced compared to the CP-μDMFC due to the lower porosity and the smaller contact angle of the rGO/CP. With increasing test concentrations, the open-circuit voltage of the rGO/CP-μDMFC decreased less compared with that of the CP-μDMFC. This was attributed to the effective reduction of the methanol crossover flux by the rGO/CP, which weakened the cathode overpotential caused by the methanol crossover.

In this work, rGO/CP was used to replace the conventional anode diffusion layer CP, which achieved a combined increase in maximum power density and an optimal methanol feeding concentration for a conventional μDMFC. The comparison to other works are listed in Table 2.

To investigate stability, the CP-μDMFC and the rGO/CP-μDMFC were placed in a 343 K thermostat and discharged at a constant current of $80\;{\mathrm{mA}/\mathrm{cm}}^{2}$. The two cells were fueled with 2 mL of their respective optimal feeding concentration (1 mol/L for the CP-μDMFC and 3 mol/L for the rGO/CP-μDMFC). The results are shown in Figure 6.

It could be seen that the discharge time of the CP-μDMFC was 140 min, while that of the rGO/CP-μDMFC was 217 min, which was 55% longer. Meanwhile, the rGO/CP-μDMFC also had a higher voltage output than the CP-μDMFC at the same current density. This indicated that the rGO/CP-μDMFC could operate stably for longer at higher powers at higher concentrations.

### 3.3. Electrochemical Impedance Spectroscopy

Considering the complex reaction process of electron transfer and matter transportation (methanol, oxygen, CO_2_, and water) in a μDMFC, the equivalent circuit model (ECM), as shown in Figure 7, was used to investigate a cell’s impedances [13,33,34]. The model used constant phase elements instead of the ideal capacitances commonly found in the traditional ECM to account for the inhomogeneous structure of the relevant electrode cross-sections.

$CPE_{i}$ and $R_{i}$ describe the capacitive behavior and contact resistance between the membrane and the catalytic layer, respectively [33]. $R_{m}$ is the ohmic resistance, $R_{mt}$ is the mass transfer resistance, $R_{ct}$ is the charge transfer resistance, and $CPE_{1}$ and $CPE_{2}$ represent the charging and discharging processes of the anode and cathode bilayer capacitance, respectively [13]. $L_{co}$ means that the current signal follows the voltage perturbation with a phase-delay due to the slowness of ${\left(CO\right)}_{ad}$ desorption, and $R_{co}$ is used to modify the phase-delay [33,34].

The ECM parameters could be identified by fitting the model to EIS curves, as shown in Figure 8.

The identified impedance parameters are listed in Table 3.

It could be seen that the $R_{i}$ were almost the same for the CP-μDMFC and the rGO/CP-μDMFC, which was reasonable since the same catalyst and Nafion membrane were used in this work [33]. The rGO/CP-μDMFC had a smaller $R_{m}$ compared to the CP-μDMFC, which was attributed to the good conductivity of rGO. The main difference in EIS was reflected in the intermediate frequency arc, which should be attributed to the anode reaction’s $R_{ct}$. By using rGO/CP as an ADL, the $R_{ct}$ of the anode reaction was reduced and the power of the rGO/CP-μDMFC was enhanced. The $R_{mt}$ of the rGO/CP-μDMFC was larger than that of the CP-μDMFC, which was attributed to its lower porosity and the smaller contact angle of rGO/CP as opposed to CP.

### 3.4. Methanol Crossover

LSV tests were performed at three different concentrations of methanol solution, 1 mol/L, 3 mol/L, and 5 mol/L, to determine the degree of methanol crossover from anode to cathode. The methanol crossover current densities for the CP-μDMFC and the rGO/CP-μDMFC are shown in Figure 9.

The peak methanol crossover current densities are shown in Table 4, with more significant peaks indicating a more severe methanol crossover.

The results showed that using rGO/CP as an ADL reduced methanol crossover. The higher the methanol concentration, the higher the percentage reduction. At a low methanol concentration of 1 mol/L, the difference in peak current density between the two cells was minor, 9.0%. When the methanol concentration increased to 3 mol/L, the difference increased to 12.8%. When the methanol concentration increased to 5 mol/L, the difference increased to 16.3%. This explained why the power difference between the rGO/CP-μDMFC and the CP-μDMFC was the largest at 5 mol/L. A higher methanol crossover had a more negative effect on cell performance. rGO/CP as an ADL could reduce methanol crossover and, thus, significantly enhance cell performance.

## 4. Conclusions

In this work, in order to enable a μDMFC to operate stably at higher methanol concentrations, a novel ADL structure with rGO/CP was prepared. The reduced graphene oxide was effectively loaded onto carbon paper by electrophoretic sedimentation combined with an in situ reduction method, resulting in a lower porosity, a smaller contact angle, and better conductivity than CP. The experimental results showed that using rGO/CP instead of CP for an ADL could enable a μDMFC to perform better. The maximum power density increased from $23.36\;{\mathrm{mW}/\mathrm{cm}}^{2}$ to $35.72\;{\mathrm{mW}/\mathrm{cm}}^{2}$, the optimal methanol feeding concentration increased from 1 mol/L to 3 mol/L, and the constant current discharge time increased from 140 min to 217 min. The EIS results showed that using rGO/CP as an ADL could effectively reduce a cell’s ohmic and charge transfer resistances. The mass transfer resistance became larger due to its lower porosity and smaller contact angle, thus reducing the methanol crossover. The LSV results confirmed that rGO/CP could reduce methanol crossover, allowing a μDMFC to operate at higher concentrations.

## Figures and Tables

**Figure 1 nanomaterials-12-02941-f001:**
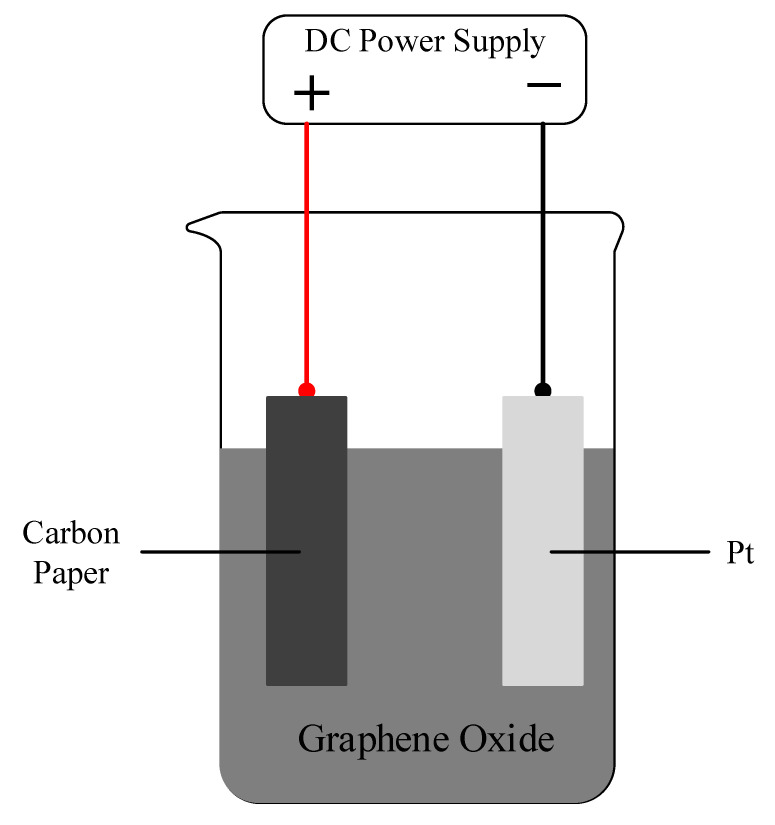
Illustration of the rGO/CP preparation.

**Figure 2 nanomaterials-12-02941-f002:**
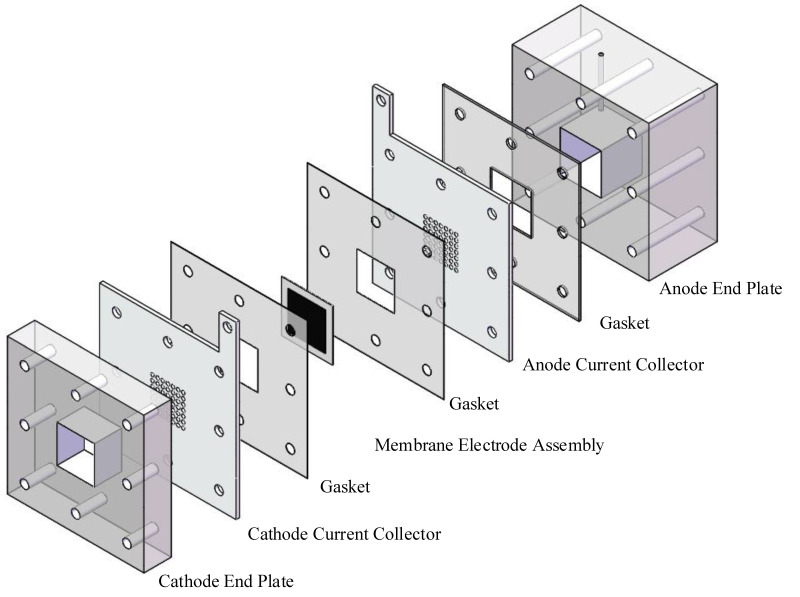
Assembled μDMFC schematic diagram.

**Figure 3 nanomaterials-12-02941-f003:**
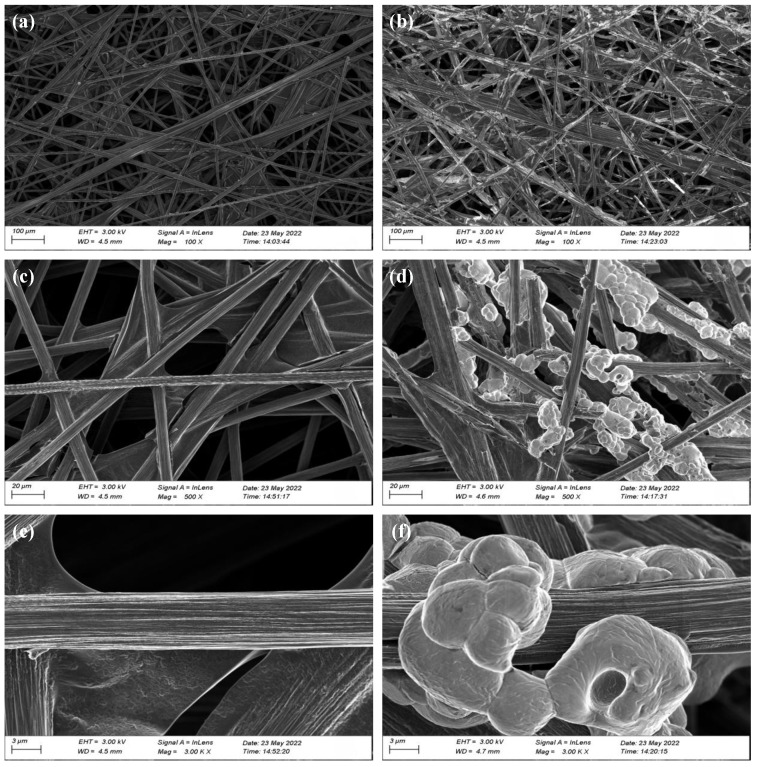
SEM images: (**a**) CP 100×; (**b**) rGO/CP 100×; (**c**) CP 500×; (**d**) rGO/CP 500×; (**e**) CP 3000×; (**f**) rGO/CP 3000×.

**Figure 4 nanomaterials-12-02941-f004:**
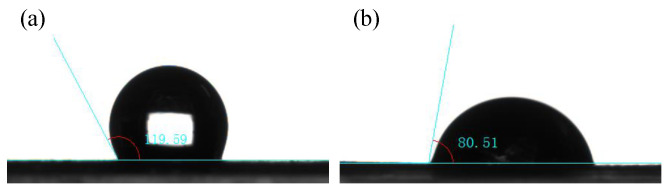
Contact angles: (**a**) CP; (**b**) rGO/CP.

**Figure 5 nanomaterials-12-02941-f005:**
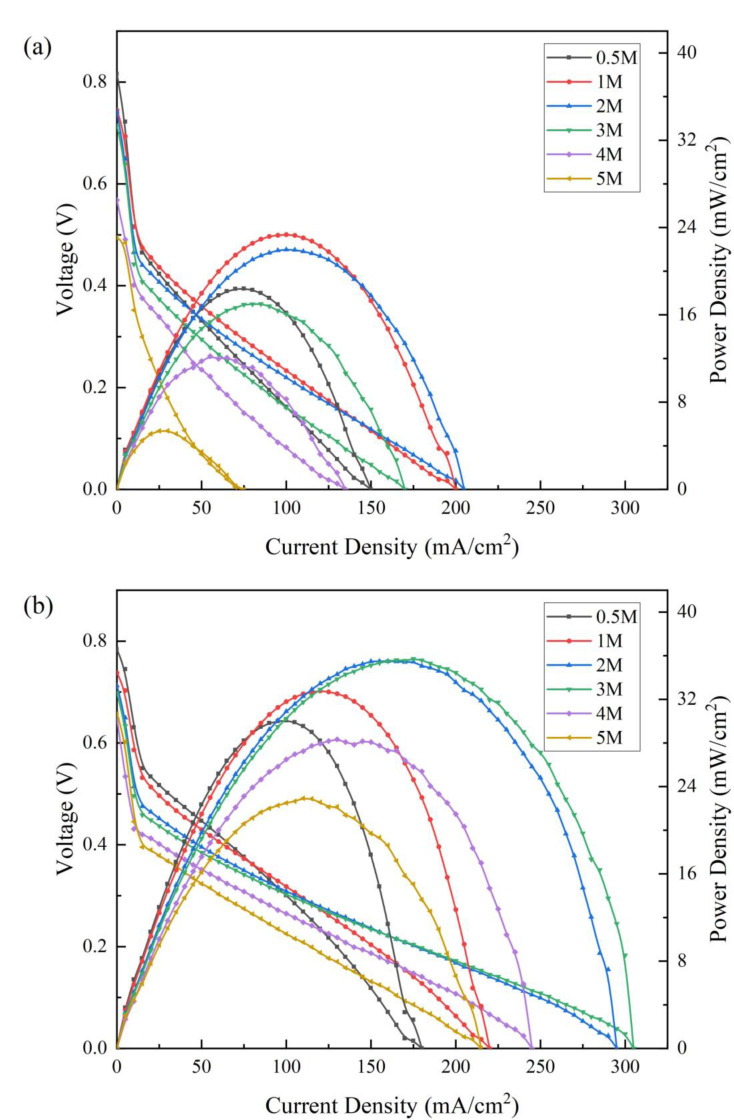
Polarization curves at different methanol concentrations at 343 K: (**a**) CP-μDMFC; (**b**) rGO/CP-μDMFC.

**Figure 6 nanomaterials-12-02941-f006:**
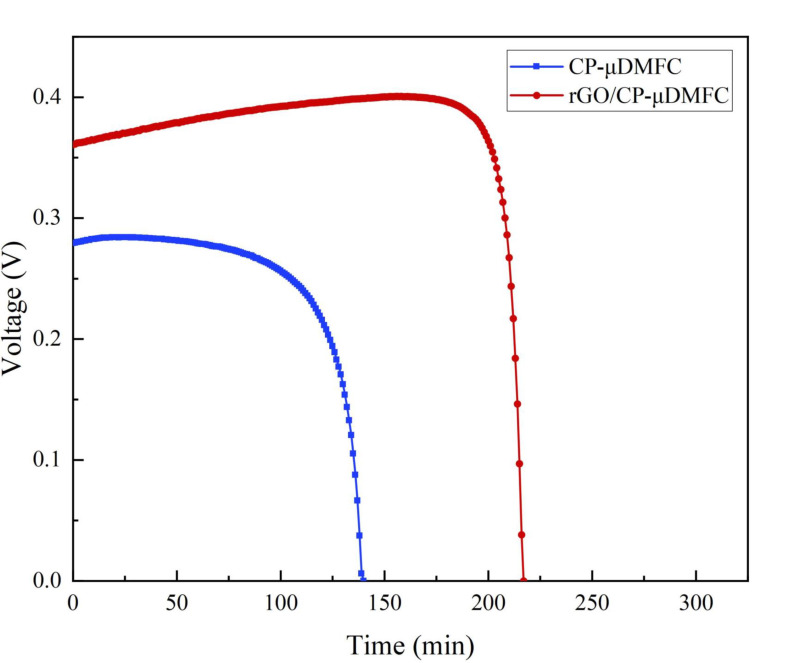
Constant current discharge curves of the CP-μDMFC and the rGO/CP-μDMFC at optimal methanol concentrations.

**Figure 7 nanomaterials-12-02941-f007:**
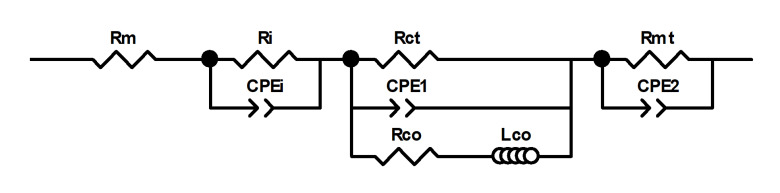
Equivalent circuit model of a μDMFC.

**Figure 8 nanomaterials-12-02941-f008:**
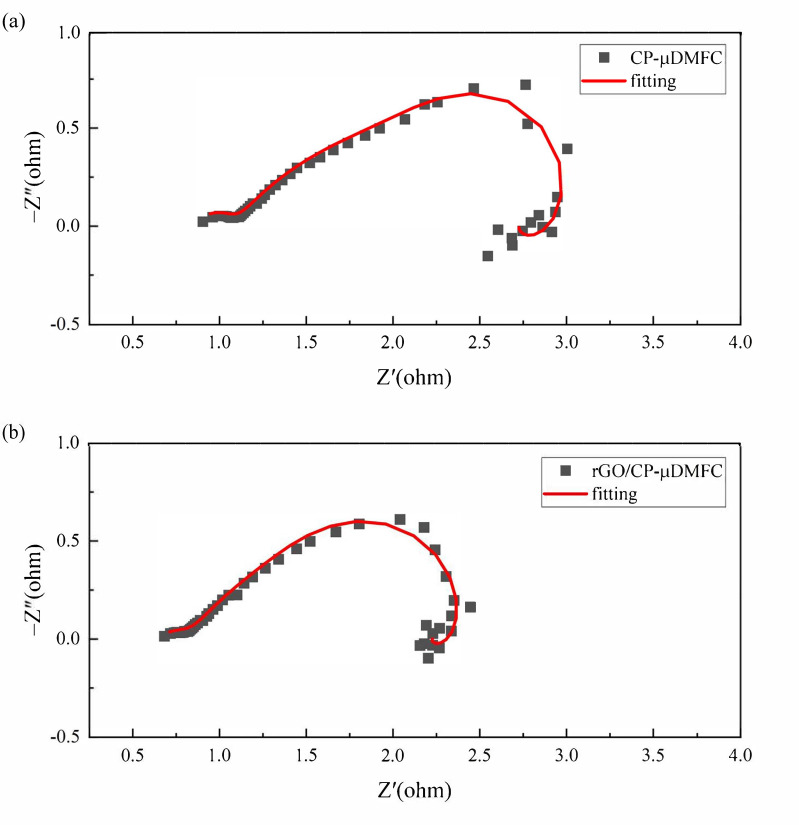
EIS fitting examples at 343 K, 2 mol/L, and $80\;{\mathrm{mA}/\mathrm{cm}}^{2}$: (**a**) CP-μDMFC; (**b**) rGO/CP-μDMFC.

**Figure 9 nanomaterials-12-02941-f009:**
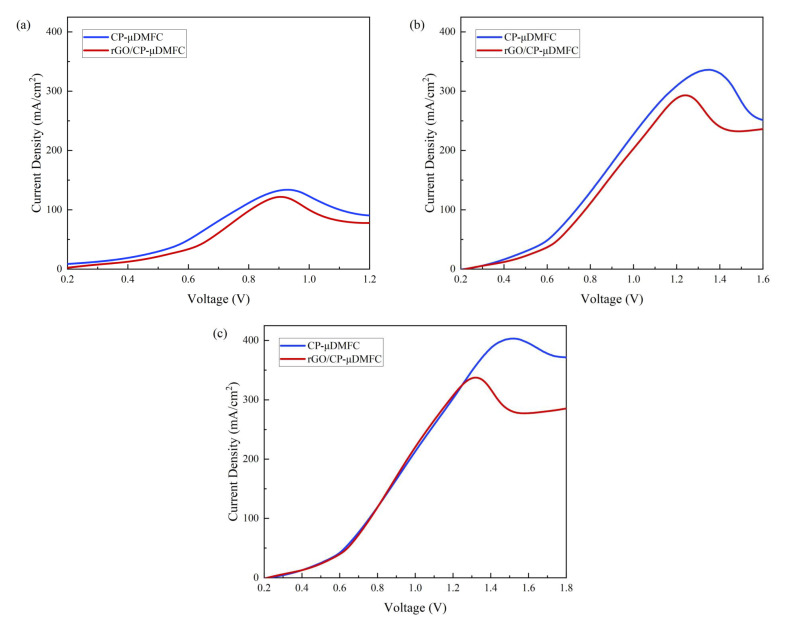
Methanol crossover current densities of the rGO/CP-μDMFC and the CP-μDMFC at different methanol concentrations: (**a**) 1 mol/L, (**b**) 3 mol/L, and (**c**) 5 mol/L.

**Table 1 nanomaterials-12-02941-t001:** Maximum power density and open-circuit voltage of the CP-μDMFC and the rGO/CP-μDMFC at different methanol concentrations.

Concentration (mol/L)	Maximum Power Density (mW/cm^2^)		Open-Circuit Voltage (V)	
	CP-μDMFC	rGO/CP-μDMFC	CP-μDMFC	rGO/CP-μDMFC
0.5	18.40	30.05	0.82	0.78
1	23.36	32.77	0.75	0.74
2	21.98	35.50	0.74	0.71
3	17.00	35.72	0.71	0.70
4	12.21	28.33	0.57	0.65
5	5.37	22.93	0.49	0.66

**Table 2 nanomaterials-12-02941-t002:** Comparison with state-of-the-art works in diffusion layer design.

Materials and Methods	P_max_ (Conventional DL)	P_max_ (Novel DL)	Reference
	(mW/cm^2^)	(mW/cm^2^)	
rGO/CP	23.36 (1 mol/L)	35.72 (3 mol/L)	This work
rGO-SSFF	31 (3 mol/L)	35 (4 mol/L)	[10]
3DG	25 (1 mol/L)	31.2 (1 mol/L)	[12]
Button-type μdmfc with 3DG DL	6.8 (1 mol/L)	9.3 (1 mol/L)	[13]
Modified CP	36.6 (6 mol/L)	36.9 (10 mol/L)	[16]
Laser-perforated CC	67.6 (2 mol/L)	89.1 (2 mol/L)	[17]
TiC/CNFs film	18.1 (2.5 mol/L)	20.2 (2.5 mol/L)	[18]
Carbonized PAN mats	nearly 85.7 (2 mol/L)	85.7 (2 mol/L)	[19]

**Table 3 nanomaterials-12-02941-t003:** Parameter identification results.

Parameters	CP-μDMFC (Ω)	rGO/CP-μDMFC (Ω)
$R_{m}$	0.848	0.634
$R_{i}$	0.248	0.242
$R_{ct}$	1.687	0.793
$R_{co}$	2.007	1.485
$R_{mt}$	0.725	0.844

**Table 4 nanomaterials-12-02941-t004:** Peak methanol crossover current densities of the rGO/CP-μDMFC and the CP-μDMFC at different methanol concentrations.

Concentration	CP-μDMFC	rGO/CP-μDMFC
(mol/L)	(${\mathrm{mA}/\mathrm{cm}}^{2}$)	(${\mathrm{mA}/\mathrm{cm}}^{2}$)
1	133.9	121.9
3	336.2	293.1
5	403.3	337.6

## Data Availability

Not applicable.

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
