# Peer review of "Reduced Graphene Oxide/Carbon Paper for the Anode Diffusion Layer of a Micro Direct Methanol Fuel Cell"

_nanomaterials, 2022, doi:10.3390/nano12172941_

Round 1

Reviewer 1 Report

The article titled "Reduced Graphene Oxide/ Carbon Paper for Anode Diffusion Layer of Micro Direct Methanol Fuel Cell" written by Zhang et al. presented a novel anode diffusion layer of reduced graphene oxide on top of carbon paper via electrophoretic sedimentation combined with in situ reduction method to improve the performance of the micro direct methanol fuel cell and to prevent methanol crossover. 

Manuscript is well written and the analytical presentation of data is good. 

I would like to know how the authors confirmed the formation of reduced graphene oxide? What kind of characterizations were performed to confirm the formation?

Also, when the carbon paper is electrophoretically deposited with graphene oxide, how did the authors confirmed if the deposited thickness is optimum for the application? Did authors tested different amounts of rGO over CP?

Also authors didn't test the performance of the rGO/CP without any catalyst to understand if the rGO is catalytically active or not towards methanol oxidation.

Reviewer 2 Report

Dear Author,

After careful reading the manuscript nanomaterials-1880417-peer-review-v1, it is clear that it details the enhanced performance of a passive micro direct methanol fuel cell, by proper modification of the carbon paper anode diffusion layer with reduced graphene oxide. Hence, in my opinion the proposed paper fits the scope of the journal, providing an innovative application for the graphene oxide reductively deposited onto carbon paper, in the field of direct methanol fuel cell devices. However, there are still some issues that need to be clarified in order to improve its scientific soundness, namely:

1-      Better use of references should be considered, to support not only the Introduction but also the explanations given along the rest of the manuscript.

2-      Regarding the Methods and Results, these can also be improved, not only by making use of the proper references, but also by detailing important information (in the text, figures and tables) now missing.

3-      Moreover, these descriptions since they are related to past events, should be given most of the times, in the past tense and not the present.

4-      Finally, some reference to how the performance of the devices prepared can be compared with others already reported in the literature, would allow further validation of the importance of this study.

Thus, I recommend that further considerations/revisions should be implemented in the manuscript in order to be acceptable for publication. More details (highlighted in yellow) can be found along the revised manuscript (pdf file in attach). I hope my comments/suggestions will help you to improve your purposed paper.

Best regards
